# INSTILLING INDUCTIVE BIASES WITH SUBNETWORKS

## ABSTRACT

Despite the recent success of artificial neural networks on a variety of tasks, we have little knowledge or control over the exact solutions these models implement. Instilling inductive biases—preferences for some solutions over others—into these models is one promising path toward understanding and controlling their behavior. Much work has been done to study the inherent inductive biases of models and instill different inductive biases through hand-designed architectures or carefully curated training regimens. In this work, we explore a more mechanistic approach: *Subtask Induction*. Our method discovers a functional subnetwork that implements a particular subtask within a trained model and uses it to instill inductive biases towards solutions utilizing that subtask. Subtask Induction is flexible and efficient, and we demonstrate its effectiveness with two experiments. First, we show that Subtask Induction significantly reduces the amount of training data required for a model to adopt a specific, generalizable solution to a modular arithmetic task. Second, we demonstrate that Subtask Induction successfully induces a human-like shape bias while increasing data efficiency for convolutional and transformer-based image classification models. Our code is available at the following *anonymous repository link*.

## 1 INTRODUCTION

Neural networks have come to dominate most fields of machine learning (He et al., 2015a; Brown et al., 2020; Radford et al., 2022; Mildenhall et al., 2020), but we have little control over the algorithms these models learn during training. To address this problem, much work has been done to instill *inductive biases* — preferences for some solutions over others — into neural networks. Studying inductive biases is interesting for at least two reasons: (1) From a practical standpoint, inductive biases could be used to discourage models from adopting solutions that leverage incorrect or biased information to make decisions (e.g. sorting job candidates on the basis of protected characteristics, or exploiting heuristics that do not generalize to a larger domain). (2) From a theoretical standpoint, human learning is thought to be mediated by a variety of inductive biases, which enable better sample efficiency and better generalization capabilities (Lake et al., 2017). Contemporary deep learning systems demonstrate weaknesses related to both of the above: they require massive datasets and computing power to train (Touvron et al., 2023; Radford et al., 2021; Dosovitskiy et al., 2020) and can often be sensitive to small perturbations of inputs (Szegedy et al., 2014; Geirhos et al., 2019; Hermann & Kornblith, 2019). Thus, a better understanding of inductive biases and how to induce them could pave the way toward improving such systems.

Current approaches to instilling inductive biases in models require either (1) limiting model expressivity through handcrafted architectural constraints, (2) metalearning over a large dataset (Griffiths et al., 2019), or (3) training or fine-tuning on augmented datasets, which may (Andreas, 2020) or may not (Huang et al., 2020; Khashabi et al., 2020) work. In contrast, we propose **Subtask Induction**, a method of instilling inductive biases by (1) localizing a subnetwork within a trained neural network that performs a specific subtask within an overall model, and (2) initializing another network with only these subnetwork weights, leaving the remaining weights randomly initialized. This instills a specific computation into a model from the outset, which provides a soft inductive bias towards solutions that leverage that subtask. We demonstrate that Subtask Induction is effective on a range of tasks and model architectures. While our results are an early proof of concept, they open a door for future research on more mechanistic approaches to instilling inductive biases. This approach is more flexible than architectural design, simpler and cheaper to train than metalearning-based approaches, and more reliable than data augmentation based approaches.

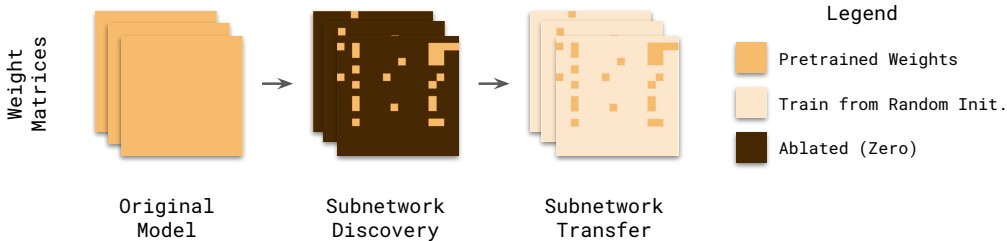

Figure 1: Subtask Induction localizes a subnetwork that implements a certain subtask in a trained neural network and transfers it to a randomly initialized model, thereby instilling an inductive bias towards solutions utilizing the specific subtask. The figure above illustrates the 3 stages of Subtask Induction in our experiments: we first train for a binary weight-level mask representing the sub-network for a specific subtask through *subnetwork discovery*, then perform *subnetwork transfer* by copying the subnetwork weights to a newly initialized model and keep it frozen while optimizing the re-initialized weights. We demonstrate through two experiments that transferring subnetworks effectively and reliably instills desired inductive biases.

Our contributions are as follows:

1. We introduce Subtask Induction, a novel method that leverages recent advancements in interpretability to instill inductive biases.

2. We demonstrate the effectiveness of Subtask Induction on an arithmetic task, showing that Subtask Induction provides a preference for learning a particular solution with minimal training signal and significantly reduces the amount of data required for generalization.

3. We generate and release Mean-pooled ImageNet, a variant of the ImageNet dataset (Russakovsky et al., 2015) where the pixel values of each image are mean-pooled within semantic segments of the image, effectively erasing local texture while retaining global shape.

4. We apply Subtask Induction to image classification on both ResNet18 and ViT models, instilling a human-like inductive bias towards classifying based on shape information, rather than texture information.

## 2 RELATED WORK

**Inductive Bias from Architectural Constraints** Imposing architectural constraints is the standard approach for instilling inductive biases in artificial neural networks. For example, convolutional neural networks (LeCun et al., 1989) and recurrent neural networks (Hochreiter & Schmidhuber, 1997; Cho et al., 2014) are both designed to exploit useful properties of their input data (i.e. shift invariance and sequential structure). Neurosymbolic approaches give even stronger inductive biases by integrating neural networks with human-designed computations, thereby limiting the kinds of solutions a model can learn (Andreas et al., 2016; Feinman & Lake, 2020; Ruis & Lake, 2022). These approaches typically perform very well in the domain that they were crafted for, but require extensive knowledge about the domain.

**Inductive Bias from Data Augmentation and Meta-learning** Data augmentation procedures have also been proposed to provide inductive biases. This approach has been validated in both vision (Geirhos et al., 2019; Hermann & Kornblith, 2019) and language (Andreas, 2020). However, the reliability of data augmentation for instilling inductive biases has been called into question (Jha et al., 2020; Huang et al., 2020; Khashabi et al., 2020). Relatedly, some work has explored a meta-learning approach toward instilling inductive biases (Griffiths et al., 2019; McCoy et al., 2019; Kumar et al., 2022; Lake, 2019). However, this approach requires meta-learning on a large dataset comprised of multiple related tasks, and the resulting model is still not guaranteed to adopt the desired inductive bias (Kumar et al., 2020).

**Mechanistic Interpretability** Our work is inspired by recent advances in *mechanistic interpretability* – a burgeoning field whose goal is to reverse engineer the algorithms that neural networks learn. Several recent works have succeeded at this goal for both toy models (Olsson et al., 2022; Nanda et al., 2023; Chughtai et al., 2023) and more realistic models (Wang et al., 2022; Hanna et al., 2023; Merullo et al., 2023). Most closely related to the present article is recent work analyzing neural networks through the lens of subnetworks (Csordás et al., 2021; Lepori et al., 2023; Casper et al., 2022; Voss et al., 2021; Hamblin et al., 2022). This line of research has shown that trained neural networks are often composed of modular subnetworks, each of which implements specific subtasks.

## 3 Localizing and Transferring Subnetworks

Subtask Induction builds upon recent work in neural network intepretability and investigates the hypothesis that one can transfer subtasks from one model to another by transferring a subnetwork encoding that information, thereby instilling an inductive bias. If this hypothesis is true, transferring a certain subtask should bias a model towards learning solutions that use that subtask. In addition, we would also expect a greater sample efficiency and faster convergence if the inductive bias turns out to be helpful to the task.

This section formalizes Subtask Induction as a two-stage process. We first localize a subnetwork within a trained model through *subnetwork discovery* (Section 3.1), which seeks to isolate a functional subtask captured by the original model. We then transfer the subnetwork (Section 3.2) to randomly initialized neural networks and train with a different objective to test if the transferred subtask provides significant inductive biases for solutions that rely on that subtask over those that do not. We provide a graphical illustration of our method in Figure 1. Our implementation is integrated with the Python package *NeuroSurgeon* (Lepori et al., 2023).

### 3.1 Localizing Subnetworks

Given a trained neural network $M_{\boldsymbol{\theta}}$ with parameters $\boldsymbol{\theta}$, we define a subnetwork as a model where a binary mask $\gamma \in \{0, 1\}^{|\boldsymbol{\theta}|}$ is applied over the original model parameters, such that $\boldsymbol{\theta}_{\text{sub}} = \boldsymbol{\theta} \odot \gamma$. In other words, a subnetwork is a variant of the original neural network where a subset of the parameters is kept the same, and the rest are set to zero. We say that a subnetwork implements a *subtask* if $M_{\boldsymbol{\theta}_{\text{sub}}}$ produces the expected outcomes of a more basic task that potentially contributes to solving the original task. E.g. a subtask for an image classification model could be a curve detector, and a subtask in a language model could be a syntax parser. If we successfully find a subnetwork that achieves a subtask, we say that such a subtask is implemented within a model.

Optimizing for a binary mask is practically intractable due to the $2^{|\boldsymbol{\theta}|}$ possible combinations. We thus apply *continuous sparsification* (Savarese et al., 2020) to train a continuous approximate of the binary mask that is discretized at test time. Continuous sparsification re-parameterizes a binary mask with element-wise sigmoid functions and schedules a scale coefficient $\beta$ that increases through training to "anneal" a soft mask to a hard one. Our implementation of this algorithm is described in more details in Appendix A. In order to find a subnetwork for a particular subtask, we train the mask by defining a new training objective that captures the subtask and perform gradient descent to localize a set of parameters that minimizes loss on the subtask. We name this process *subnetwork discovery*.

### 3.2 Transferring Subnetworks

After obtaining a subnetwork with mask $\gamma_{\text{sub}}$, we initiate a *subnetwork transfer* by transferring the parameters within the subnetwork (i.e. parameters where $\gamma_{\text{sub }i} = 1$) to a randomly initialized copy of the model. We then train the network on the new training objective. During training, we only optimize the randomly initialized parameters and keep the subnetwork frozen.

Let $\mathcal{L}_{\text{new}}$ denote the optimization objective of the new task, $\boldsymbol{\theta}_{\text{original}}$ denote pretrained parameters, and $\boldsymbol{\theta}_{\text{new}}$ denote the re-initialized parameters. The training objective then becomes

$$\operatorname*{argmin}_{\boldsymbol{\theta}_{\text{new}} \in \mathbb{R}^{|\boldsymbol{\theta}|}} \left( \mathcal{L}_{\text{new}} \left( M_{\gamma_{\text{sub}} \odot \boldsymbol{\theta}_{\text{original}} + (1 - \gamma_{\text{sub}}) \odot \boldsymbol{\theta}_{\text{new}}} \right) \right). \tag{1}$$

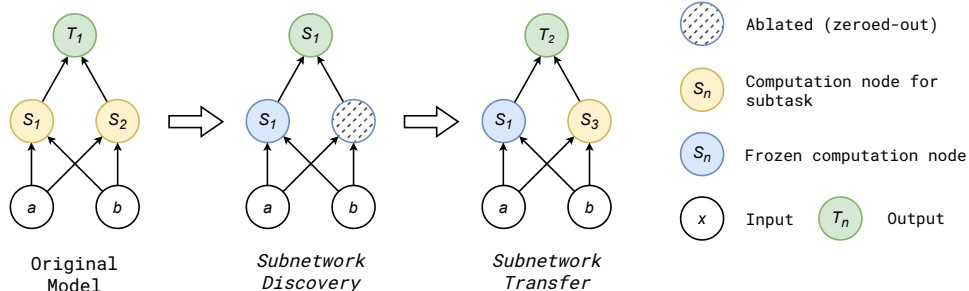

Figure 2: Graphical illustration of our experimental setup. Tasks $T_1$ and $T_2$ are setup to be combinations of three subtasks, $S_1$, $S_2$, and $S_3$, where $S_1$ is shared between the two. We train a model on $T_1$, then perform Subtask Induction by localizing and transferring the shared subtask $S_1$ to instill inductive biases towards a new model trained on $T_2$. We find that transferring the subnetwork improves the model's ability to learn $T_2$ significantly.

## 4 ARITHMETIC EXPERIMENTS

To verify the effectiveness of Subtask Induction, we train neural networks on an arithmetic dataset, where subtasks can be easily defined and tested. For this, we use tasks in the form of those studied by Power et al. (2022). In Power et al.'s experiments, an overparameterized neural network is trained on a synthetic dataset of some computation $a \circ b = c$, where $a, b$, and $c$ are discrete symbols and $\circ$ denotes an arithmetic operation with two arguments (for example, $a + b$ or $a^2 + ab$). We isolate a subnetwork implementing some particular subtask of the original training task. We then transfer this subnetwork to a new task that should benefit from having access to this subtask.

### 4.1 DATASET

We algorithmically generate datasets by defining a computation $\circ$ and sampling two integers $a$ and $b$ from a chosen range $[\,0, \max)$. We then formulate the expression into a sequence of four tokens `<a>  <sep> <c>` where each element in a pair of brackets indicates a token. Here "sep" represents the special separator token, and $c$ is the expected output of the computation $a \circ b$. This formulation allows us to train a decoder-only transformer on the sequence with a standard next-token prediction objective.

In all of the following experiments, we fix max = 1000. We tokenize each number into a discrete symbolic token, rather than an integer or floating point representation, and each token embedding is learned individually. Since each number is represented by a discrete token, we constrain the dataset such that each of the possible tokens must appear at least once in the training set. Following prior work (Power et al., 2022; Nanda et al., 2023), we take modulo of the output by a prime number $p$ to restrict the output space (i.e. the operation is always in the form "$a \star b \,(\mathrm{mod}\ p)$", where the modulo is taken after the two-place operator "$\star$"). In all our experiments we fix $p = 7$.

### 4.2 EXPERIMENTAL SETUP

We generate training data for two tasks, $T_1 := a + ab \,(\mathrm{mod}\ p)$ and $T_2 := a^2 + ab \,(\mathrm{mod}\ p)$. Note that the two tasks can be described as the combination of results from subtasks $S_1 := ab \,(\mathrm{mod}\ p)$, $S_2 := a \,(\mathrm{mod}\ p)$, $S_3 := a^2 \,(\mathrm{mod}\ p)$, and $T_1$ and $T_2$ share the computation node $S_1$. We perform Subtask Induction from $T_1$ to $T_2$ by transferring $S_1$. Figure 2 demonstrates this procedure graphically. The experiment follows three steps:

1. Train a neural network on $T_1$, where it is expected to solve an arithmetic task.
2. Performing subnetwork discovery to localize a subnetwork that solves $S_1$.
3. Transferring the subnetwork to $T_2$ and test for inductive bias towards solutions utilizing $S_1$.

In step 1, we generate training data for the computation $T_1$ by randomly sampling 20% of the total $1000^2$ combinations, which gives us 200,000 rows of training data. We use another independently

generated set of 20,000 samples for test data. We train a decoder-only transformer on this dataset with a standard next token prediction objective, and report accuracy/loss on the last token, as the last token represents the solution to the problem.

This task $a + ab \pmod{p}$ can be intuitively broken down into constituent subroutines: computing $a \pmod{p}$, computing $ab \pmod{p}$, and combining the results into the final output. We hypothesize that models also implicitly decompose the task in this manner. To probe for a subroutine responsible for the computation $ab \pmod{p}$, we generate 50,000 samples of the computation $ab \pmod{p}$, and perform *subnetwork discovery*. This step gives us a binary mask $\gamma_{sub}$, and the subnetwork $M_{\boldsymbol{\theta} \odot \gamma_{sub}}$ should perform the computation $ab \pmod{p}$ instead of the original training objective $a + ab \pmod{p}$.

We then investigate if the subnetwork provides an inductive bias toward a solution utilizing the subtask. We intentionally make the training objective $T_2$ appear ambiguous by supplying the model a minimal dataset of 1000 samples of the format $\mathbb{X}_i^{n=1000} = $ `, , <sep>, <i ∘ i>`, where the two inputs are identical. This ensures that each discrete token has appeared at least once while leaving the training task ambiguous. Concretely, the objective would be ambiguous between computations $2a^2 \pmod{p}$, $2b^2 \pmod{p}$, and $a^2 + b^2 \pmod{p}$.

In addition to the minimal dataset above, we manipulate the number of *disambiguation samples* present in the training set, i.e., training examples in which the two inputs are no longer constrained to be identical. These are randomly sampled from the input space of $\{0, 1, 2, ..., 999\}^2$, and provides information to disambiguate the correct computation $T_2$ from other possible computations.

We vary the number of disambiguation samples to quantify the inductive bias of neural networks. With a strong inductive bias towards the correct rule, a small number of disambiguating examples would be enough to disambiguate the task[1]. If Subtask Induction is effective, it should enable the model to achieve higher accuracies with fewer disambiguating examples. The evaluation set and the test set always contain 1000 data points, each of which is generated independently from a random sample over all possible combinations.

We experiment with several GPT2 configurations, varying the number of layers from 2 to 12. We vary the number of disambiguation samples from 10 to $10^4$ (0.001% to 1% of total possible combinations, respectively) with constant intervals for a total of 16 different sample sizes on each model. After transferring subnetwork weights, we train each model for 100 epochs and save the model with best accuracy on the evaluation set, and then report the accuracy achieved on the test set (See Appendix B.1 for model configuration and training details).

### 4.3 RESULTS

If Subtask Induction successfully instills an inductive bias, we would expect our model to achieve higher test accuracy with less training data, relative to a randomly initialized model. We find this to be the case: as shown in Figure 3, models initialized with subnetworks with as few as 3.2% of total parameters (see Table 2) representing subtask $S_1$ gain significant inductive bias towards the solution utilizing $S_1$. This is evidenced by the significantly higher sample efficiency: all model configurations trained with Subtask Induction achieve near-perfect accuracy with as few as 1000 disambiguation training samples (0.1% of total possible combinations). As a comparison, models trained from scratch only average 50.6% test accuracy when trained on the same data and never reach perfect generalization accuracy within the range of training samples tested (0 to $10^4$).

We set up the following controls to validate the effectiveness of Subtask Induction:

1. Comparison with full model transfer: Since the subnetwork captures $S_1$, the only shared computation between $T_1$ and $T_2$, we hypothesize that it carries all the "helpful" information a neural network trained on $T_1$ could provide, and thus expect Subtask Induction to have comparable performance as transferring the entire model trained on $T_1$. This turns out to be the case: Across sample sizes and model configurations, transferring subnetworks of around 3% to 7% parameters achieves at least as good generalization accuracy and sample efficiency as transferring the entire model.

---

[1]Ideally, with a sufficiently strong inductive bias, no unambiguous examples would be required, though in practice we do not obtain such a strong inductive bias.

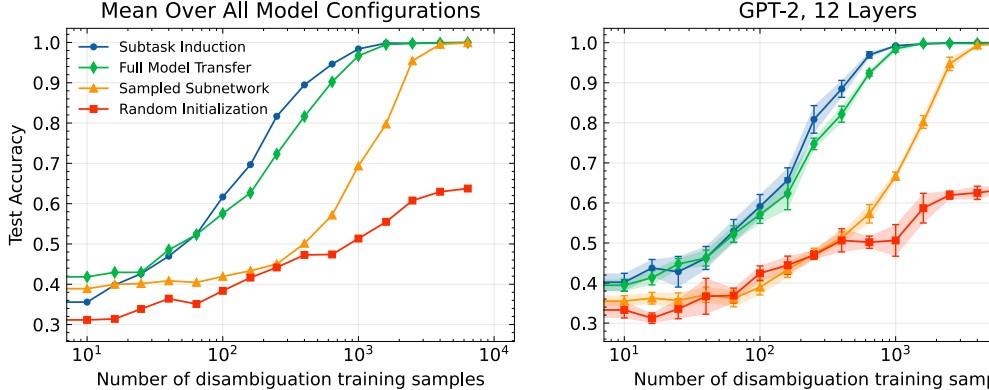

Figure 3: Test accuracy vs number of disambiguation training samples. Left: average over all model configurations (GPT-2, 2 to 12 layers), right: One configuration (GPT-2, 12 layers) with standard deviation across 5 runs. The horizontal axis is in log scale. Trials shown in Figure include Subtask Induction compared against 3 controls: randomly initialized model, transferring randomly sampled subnetworks, and transferring the entire model trained on $T_1$. Despite transferring less than 10% of all parameters, Subtask Induction yields comparable and often higher accuracy compared to transferring the entire model and boosts data efficiency significantly compared to random controls.

2. Comparison with randomly sampled subnetwork: Intuitively, transferring a subset of parameters from a model trained on $T_1$ could provide benefits for training on $T_2$ purely due to the similarity of the two tasks. We control for this by sampling a random subnetwork containing the same number of parameters as a subnetwork localized through subnetwork discovery[2] and transferring the sampled subnetwork. This gives uniformly worse results: while still better than random initialization, a randomly sampled subnetwork requires on average around 6 times as much data in order to reach perfect generalization accuracy.

In addition to the results in Figure 3, all of the patterns reported above hold in each of the individual model configurations as well. We also experiment with a range of different arithmetic tasks (e.g. $a^3 + ab$) and subnetworks. We report these extended results and additional analysis in Appendix B.

## 5 VISION EXPERIMENTS

In this section we apply Subtask Induction on image classification tasks, a highly complex domain for which no complete algorithmic solutions are known. While contemporary deep neural networks are able to meet or even exceed human-level accuracy on image classification (He et al., 2015b;a; Dosovitskiy et al., 2020), they often rely on a very different set of cues than humans do, thereby limiting their robustness and generalization capabilities (Dodge & Karam, 2017). Prominently, while human learners overwhelmingly rely on shape information (Landau et al., 1988), convolutional neural networks are primarily reliant on local texture (Geirhos et al., 2019). We show that by localizing and transferring subnetworks within pretrained models, it is possible to instill a more human-like bias towards shape information.

### 5.1 DATASET: MEAN-POOLED IMAGENET

In order to quantify the shape and texture biases of image classification models, we introduce Mean-pooled ImageNet, a variant of ImageNet where local, high-frequency texture information of images is removed while maintaining global shape information. We use Segment Anything (Kirillov et al.,

---

[2]To ensure as fair a comparison as possible, the randomly sampled subnetwork is sampled over the same layers as the subnetwork (i.e. all the attention layers and feed-forward MLPs, but not the embedding layers), and the number of parameters sampled at each individual layer is controlled to be the same as the trained subnetwork on the respective layer. This eliminates possibilities that simply sampling the right number of parameters per layer gives equivalent results.

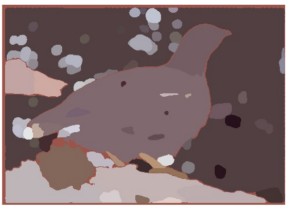 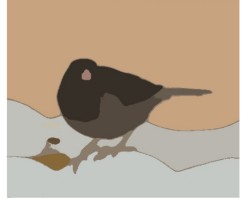 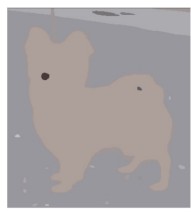 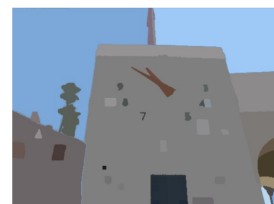

Figure 4: Qualitative evaluation of Mean-pooled ImageNet. Semantic segmentation followed by mean pooling retains most shape information in a naturalistic way while erasing local texture.

2023) to partition the image into semantic segments. After obtaining an image embedding, we query each image with a 16×16 grid of points to obtain semantic segments corresponding to each segment. To ensure that small but semantically relevant patches are not missed by the initial sampling, we further query on a 2×2 crop of the image and collect masks returned by the query. We then filter out masks that are smaller than 100 pixels and combine all masks for a non overlapping set of segments covering the entire image. Lastly, we replace each pixel value in the image by the mean pixel value of the segment it belongs to. We provide a few samples of Mean-pooled ImageNet for qualitative evaluation in Figure 4 and invite the reader to guess their corresponding classes.

Mean-pooled ImageNet employs a naturalistic augmentation strategy as it does not shift the overall color scheme of images or intentionally occlude any information apart from local texture. For humans, this augmentation is unlikely to dramatically raise the difficulty of the task or impact a classification decision. However, we find this dataset to be challenging for image classfication models. While ResNet18 reaches 95.4% accuracy when fine-tuned on 16-class ImageNet, its accuracy on the mean-pooled counterpart is only 36.8%. ViT performs much better on this dataset, but still only achieves 57.3% accuracy.

## 5.2 EXPERIMENTAL SETUP

Similar to the experiments on arithmetic tasks, we instill different inductive biases into image classification models by localizing a subnetwork within a pretrained image classification model using Mean-pooled ImageNet, and then transferring the subnetwork into a re-initialized model.

We perform all our experiments on 16-class ImageNet (Geirhos et al., 2019) and its mean-pooled counterpart. Each class label is aggregated from one or multiple ImageNet classes. The dataset contains a total of 213k images from 16 common classes in the train split of ImageNet. As the dataset is unbalanced between classes, we additionally create two smaller but class-balanced subsets: a total of 13.9k randomly downsampled mean-pooled images are used to discover the subnetwork within a pretrained model, and an additional 1.54k images are used for evaluation and model selection. We de-duplicate our evaluation dataset with our training datasets and report accuracy on the validation split of ImageNet, which is not used for either training or model selection.

We experiment with two model architectures: ResNet18 (He et al., 2015a) and ViT-base (Dosovitskiy et al., 2020). We perform subnetwork discovery on both models to localize a subnetwork that maximizes accuracy on mean-pooled images. Lastly, we transfer the subnetwork weights and re-train the model on 16-class ImageNet. As baselines, we compare against pretrained models that are finetuned on a data mixture of 213K 16-class ImageNet images *and* 15.4K mean-pooled images. This approach mimics the data augmentation approach to instilling inductive biases that have been explored in prior work (Andreas, 2020). We also compare against training these models from scratch and fine-tuning only the classification head of base models, which quantifies inherent inductive bias of the architecture and the performance base models, respectively.

## 5.3 RESULTS

**Pretrained Models Capture Shape Subtasks** For both ResNet18 and ViT, we are able to discover subnetworks achieving significantly higher accuracy on mean-pooled images than the original model, suggesting that shape-reliant subtasks exist within the original model. Within ResNet18, we find a subnetwork with 14.9% of the parameters achieving 73.8% classification accuracies on

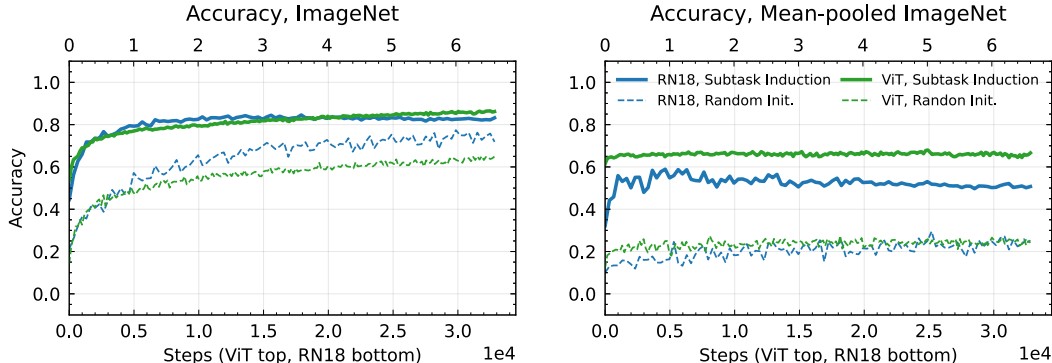

Figure 5: Training dynamics Comparison, Subtask Induction and training from scratch for ResNet18 and ViT. Left: evaluation accuracy on original ImageNet images, right: evaluation accuracy on Mean-Pooled Imagenet. Models initialized with Subtask Induction reach higher accuracies with fewer optimization steps and retain a much higher accuracy on Mean-pooled ImageNet.

Table 1: Test accuracy of Subtask Induction compared with other training strategies. We note that: (1) Subtask Induction instills a strong shape bias (18.8% performance increase on Mean-pooled ImageNet for ResNet18, 8.7% for ViT) despite the re-initialized network never being directly trained on mean-pooled images, while data augmentation does not provide such bias, (2) Subtask Induction increases sample efficiency, as both ResNet and ViT reach much higher accuracy compared to from-scratch models when trained on 16-class ImageNet, (3) Subtask Induction gives much more robust models as seen on the Cue Conflict results, where our ResNet18 outperforms pretrained ResNet18 and reaches levels comparable to pretrained ViT. While ViT trained with Subtask Induction is not as strong, it still performs significantly better than data mixture and from-scratch baselines and has the best performance on mean-pooled images.

| Model | Train Set Size | ImageNet | | Cue Conflict | |
| | | Original | Pooled | Accuracy | Robustness |
| --- | --- | --- | --- | --- | --- |
| RN18 + Subtask Induction | 213k | 80.7% | **55.6%** | **27.1%** | **77.4%** |
| RN18 from scratch | 213k | 68.9% | 24.7 % | 15.9% | 75.3% |
| RN18 + Data Mixture | 1.28M + 15.4k[1] | 91.9% | 38.3% | 18.9% | 55.3% |
| RN18 Pretrained | 1.28M | **95.4%** | 36.8% | 18.9% | 56.0% |
| ViT + Subtask Induction | 213k | 83.4% | **66.0%** | 20.0% | 72.1% |
| ViT from scratch | 213k | 58.4% | 23.4% | 12.1% | 70.3% |
| ViT + Data Mixture | 14.2M + 15.4k[1] | 84.3% | 35.1% | 15.0% | 64.7% |
| ViT Pretrained | 14.2M | **97.1%** | 57.3% | **28.5%** | **73.8%** |

[1] Data Mixture refers to the fine-tuning of a pretrained model with a mixture of original images and additional mean-pooled images (the same 15.4k used for subnetwork discovery) in order to instill a bias towards shape-based classification

mean-pooled images. In ViT, we were able to localize a 14.6% parameter subnetwork achieving 76.1% accuracy on mean-pooled ImageNet. Both achieve a significant accuracy boost compared to pretrained models.

**Subtask Induction Increases Sample Efficiency** In Figure 5, we show the training dynamics of ResNet and ViT trained with Subtask Induction compared against those trained from random initialization. We see that models initialized from subnetworks are much more data and computation efficient: on ResNet18, we observe that it achieves 11.8% better accuracy when trained on the same dataset; ViT proves to be much more data hungry as it fails to achieve competitive accuracies when trained on the 213k images of 16-class ImageNet. We also observe that the performance on mean-pooled images is maintained throughout training, suggesting that solutions learned by both models rely on the transferred subtask. In comparison, models trained from scratch with our small dataset do not generalize to mean-pooled images.

**Transferring Subnetworks Instills Stronger Shape Bias**    We present results of Subtask Induction compared against various baselines in Table 1. When the subnetworks are transferred and re-trained on 16-class ImageNet, we find that they achieve competitive accuracies on the original images and significantly better accuracies on mean-pooled images, suggesting a much stronger shape bias. In comparison, fine-tuning pretrained models and training from scratch with the mean-pooled data augmentation both fail to generalize to the held-out mean-pooled images. Notably, we show that Subtask Induction successfully instills a shape bias into ResNet, allowing it to achieve an accuracy comparable to pretrained ViT and 18.8% better than pretrained ResNet18, all while being trained on a much smaller dataset (17% and 1.5% the size of ResNet and ViT training set, respectively). While Subtask Induction gives weaker performance boosts to ViT, it still increases accuracy on mean-pooled images by 8.7% and performs much better in every benchmark compared to training a model from scratch on the same dataset.

In addition, we also observe that fine-tuning the model with data augmentation achieves uniformly worse overall accuracy compared to using the pretrained model and only adapting the classifier layer, suggesting that the small mean-pooled dataset used for subnetwork discovery does not give model a shape bias when used for fine-tuning. This resonates with the finding in Jha et al. (2020): when a model is finetuned on a small out-of-domain dataset, data augmentation often hurts especially if the useful information in augmented data is hard to extract.

## 5.4    ANALYSIS: CUE CONFLICT

Next, we evaluate all of our models on the cue-conflict dataset introduced in Geirhos et al. (2019), a dataset consisting of images in which texture and shape cues are dissociated from one another. For example, this dataset contains images of dogs with the texture of an elephant overlaid on them. Cue-conflict images attempt to exploit a model's texture bias to change their prediction. For each model, we report two metrics: (1) *accuracy* is the proportion of cue-conflict images that are classified correctly according to shape cues, (2) *robustness* is the proportion of image that are *not* classified according to misleading texture cues. Ideally we would want a model to achieve high performance on *both* accuracy and robustness. From the Cue Conflict columns of Table 1, we see that Subtask Induction consistently yields more accurate and robust models than fine-tuning with data augmentation. Consistent with our ImageNet results, we find that pretrained ViT already has a strong shape bias. However, it was also trained on orders of magnitude more data (14.2M vs 213k) than our ViT with Subtask Induction, which achieves comparable robustness on the cue-conflict data. Importantly, we also find that ResNet-18 trained with Subtask Induction achieves similar level of accuracy and robustness as pre-trained ViT, despite the small amount of training data and the inherent texture inductive bias of the ResNet architecture.

## 6    DISCUSSION

Inductive biases are crucial for understanding and controlling the solutions neural networks learn. We present a new technique, Subtask Induction, that leverages recent advances in our mechanistic understanding of trained models to instill such biases in neural networks. Across a range of experimental settings and model architectures, we demonstrated that Subtask Induction consistently confers the inductive bias that we expect, yielding increased sample efficiency and robustness to out of distribution stimuli. Furthermore, we demonstrated that our method has higher sample efficiency and outperforms data augmentation approaches to instilling inductive biases.

**Future Work**    Subtask Induction can be applied in wider contexts to instill specific inductive biases, either to encourage a model to learn particular solutions under limited data settings or to combat existing model heuristics. Though Subtask Induction is promising, we also note several limitations and avenues for future work. First, subtask induction requires supervised training of a binary mask to perform subnetwork discovery, which requires constructing custom-designed datasets. Future work might relax this constraint by decomposing a trained model in an unsupervised fashion, and transferring subnetworks that are discovered by this decomposition. Furthermore, Subtask Induction directly transfers subnetworks, which is only possible between models of identical architecture. Future work might seek to address this, perhaps by combining Subtask Induction with methods for re-scaling models, such as the Linear Growth Operator (Wang et al., 2023).

## 7 ETHICS STATEMENT

We believe that the present work is in compliance with the ICLR code of ethics. Subtask induction can be used to influence the solutions that neural networks learn. This may have future implications for bias, fairness, and safety of neural network models. However, we emphasize that the current iteration of subtask induction is a proof of concept, and cannot and should not be used to render models free from social biases in real-world systems.

## 8 REPRODUCIBILITY STATEMENT

To facilitate reproducibility, we provide detailed description of the models and training details in both the main text and the appendix. Specifically, the experimental setup section in both the arithmetic experiments (Section 4.2) and the vision experiments (Section 5.2) describes the configurations of our model and the baselines. We use the official released weights from original authors for ViT-base and ResNet18 for subnetwork discovery in Section 5. In addition, detailed explanation for our hyperparameters and hyperparameter search strategies if applicable are provided in Appendix B.1 and C.1. We release original code and configuration files.

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

## A    Continuous Sparsification for Subnetwork Discovery

For a trained neural network $M_{\boldsymbol{\theta}}$ with parameters $\boldsymbol{\theta}$ and a weight-level binary mask $\gamma \in \mathbb{R}^{|\boldsymbol{\theta}|}$, continuous sparsification re-parameterizes the binary mask with mask weights $\boldsymbol{s} \in \mathbb{R}^{|\boldsymbol{\theta}|}$ as $\gamma = \sigma(\beta \boldsymbol{s})$, where $\sigma$ stands for the sigmoid function $\frac{1}{1+e^{-x}}$ and $\beta$ is a scalar value scheduled in training. Notice that this allows us to compute the gradients of $\boldsymbol{s}$ and use it to update the mask values. In addition, we recover a hard binary mask as $\beta \to \infty$.

During training, we start with $\beta_{\text{start}} = 1$ and apply an exponential scheduler such that it increases by a fixed percentage each epoch until it reaches $\beta_{\text{final}}$. By setting $\beta_{\text{final}}$ to a large value, we obtain a mask that "anneals" from a soft sigmoid mask to an approximate of a binary mask at the end of training. At test time we use a binary mask $\mathbb{1}_{\boldsymbol{s}>0}$ instead, which is equivalent to setting $\beta$ to infinity.

For subnetworks with similar performance, we wish to find ones that are more sparse in order to minimize the inclusion of noise and unrelated parameters. We achieve this by adding an l0 penalty term based on the number of non-zero elements in the mask $\lambda \cdot \|\gamma\|_0$ . Notice that under continuous sparsification, this is equivalent to a ReLU-like penalty on the mask weights.

In order to localize a subnetwork achieving some specific subtask, we define a new training objective, which potentially includes new training data, different output space, and different training objectives. Let $\mathcal{L}_{\text{sub}}$ be an objective function defined for a subroutine (e.g. cross-entropy loss on input-output pairs), localizing the subnetwork essentially becomes solving

$$\underset{\boldsymbol{s}\in\mathbb{R}^{|\boldsymbol{\theta}|}}{\operatorname{argmin}} \left( \mathcal{L}_{\text{sub}} \left( M_{\boldsymbol{\theta} \odot \sigma(\beta \boldsymbol{s})} \right) + \lambda \cdot \| \sigma(\beta \boldsymbol{s})\|_0 \right). \tag{2}$$

Algorithm 1 describes our implementation. We fix $\beta_{\text{final}} = 100$ and initialize mask weights as $-0.1$ for all our experiments.

---

**Algorithm 1** Subnetwork Discovery through Continuous Sparsification

---

**Require:** model $M_{\boldsymbol{\theta}}$ with trained parameters $\boldsymbol{\theta}$, final scale coefficient $\beta_{\text{final}}$, mask initialization value $\alpha$, l0 penalty $\lambda$.

    $\beta \leftarrow 1$
    $\boldsymbol{s} \leftarrow \alpha^{|\boldsymbol{\theta}|}$
    epoch $\leftarrow n$
    **while** epoch $\neq 0$ **do**
        Forward pass on $M_{\boldsymbol{\theta} \odot \sigma(\beta \boldsymbol{s})}$
        loss $\leftarrow$ loss $+ \lambda \cdot \|\sigma(\beta \boldsymbol{s})\|_0$
        Gradient descent on loss to update $\boldsymbol{s}$              $\triangleright$ Note that $\boldsymbol{\theta}$ is not updated
        $\beta \leftarrow \beta \cdot (\beta_{\text{final}})^{\frac{1}{n}}$         $\triangleright$ Exponential scheduler for $\beta$ increases it at each epoch end
        epoch $\leftarrow$ epoch $-1$
    **end while**
        **return** $\gamma = \mathbb{1}_{\boldsymbol{s} > 0}$

---

# B  EXTENDED ARITHMETIC TASK RESULTS

## B.1  MODEL CONFIGURATION AND HYPERPARAMETERS

For the original task $T_1$, we train decoder-only transformers of the GPT-2 architecture (Radford et al., 2019) with 4 attention heads, embedding size of 128, and intermediate representation size of 512. To increase the diversity of models, we repeat for a range of layer counts $\{2, 4, 6, 8, 10, 12\}$. When trained with AdamW optimizer with $\beta_1 = 0.9, \beta_2 = 0.999$, and a learning rate of $5^{-4}$, all model configurations successfully learn the computation, as evidenced by the near-perfect or perfect ($\geq 0.999$) accuracy on the independent test set.

We localize the subnetwork by freezing all original parameters and initializing a mask $\gamma_{\text{sub}}$ over every attention and feed-foward layer of the original model. We train the mask for 50 epochs with $\beta_{\text{final}} = 100$. We experiment with sparsity penalty $\lambda \in \{1.0^{-7}, 5.0^{-7}, 1.0^{-8}\}$ and learning rate $1.0^{-3}$, and obtain subnetworks of different sparsity levels for the subroutine $S_1$. All subnetworks achieve perfect or near-perfect accuracy on the test set.

During subnetwork transfer, we search for learning rate on a sparser subset of different sample sizes for each architecture within the range $\{2^{-3}, 1^{-3}, 5^{-4}, 2^{-4}, 1^{-4}\}$, and find $2^{-4}$ to be the best performing on validation set across the board. We thus use a learning rate of $2^{-4}$ for our final runs.

## B.2  COMPLETE RESULTS BY MODEL CONFIGURATION

We report the individual results for different model configurations presented in the averaged plot in Figure 3. We also report the percentage of parameters transferred for each model configuration in Table 2. In addition to sampling a random subnetwork of the same size and transferring it as control, we also run trials where we sample a random subnetwork from the complement. Both random controls achieve similar performance and are significantly worse than Subtask Induction.

| Model configuration (# of layers) ‖ | 2 | 4 | 6 | 8 | 10 | 12 |
|---|---|---|---|---|---|---|
| Percentage of parameters (%) ‖ | 7.68% | 7.00% | 4.70% | 3.55% | 3.56% | 3.20% |

Table 2: Percentage of parameters in transferred constituent subnetworks in Figure 3. Value indicates overall percentage within masked layers, which includes attention and feed-forward layers, but not the embedding layer. We find that larger models generally have smaller subnetworks when trained on the same setup.

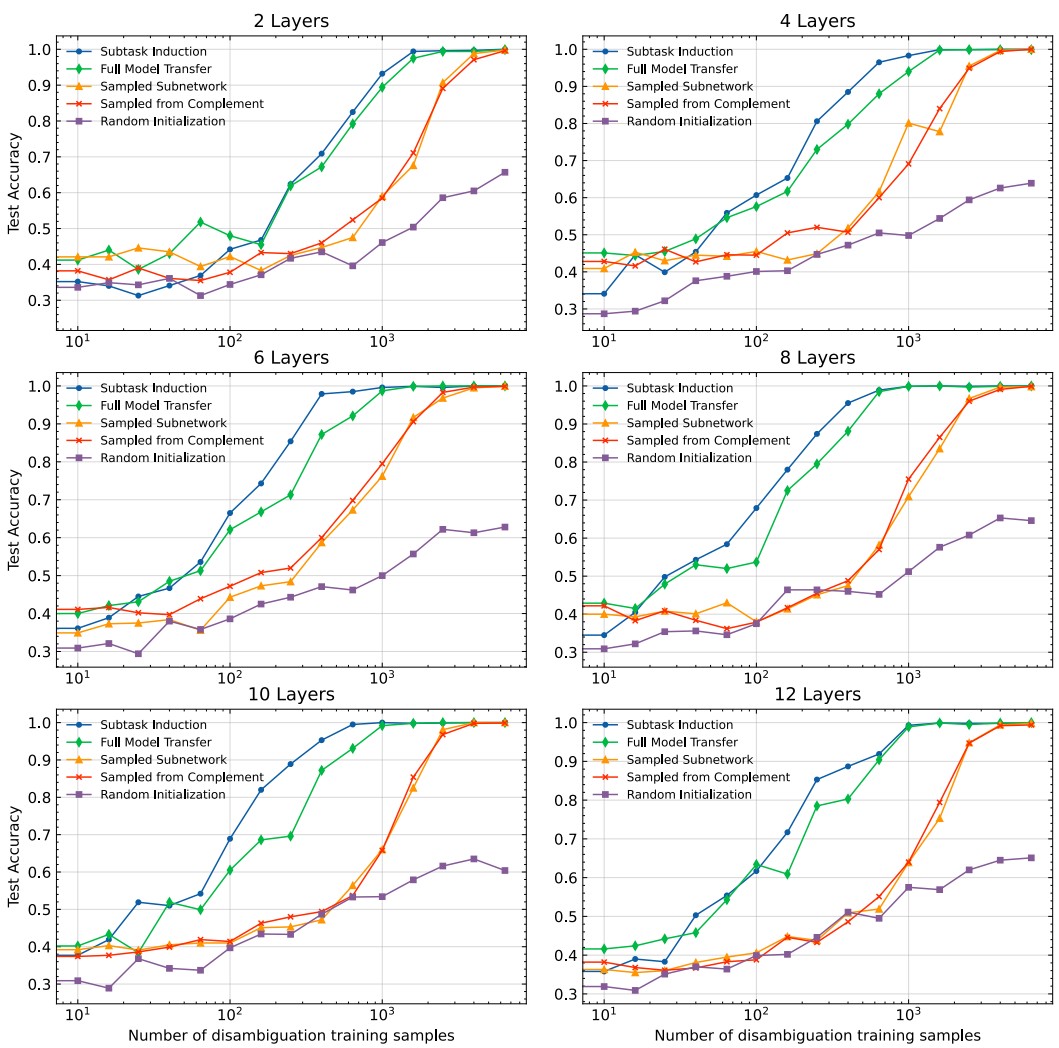

Figure 6: Test accuracy vs number of disambiguation training samples, breakdown of results of each model shown in Figure 3. The effect of Subtask Induction is constant across different number of layers, suggesting that the inductive bias the model gains does not seem to depend on depth of the model, at least in this arithmetic task.

### B.3 EFFECT OF GRADIENT UPDATES WITHIN THE SUBNETWORK

Figure 7 demonstrates a comparison between allowing and not allowing gradient updates for the subnetwork transferred to the re-initialized model. Surprisingly, allowing for gradient updates within a sampled subnetwork brings its performance up to a level comparable with transferring the entire model, but still not as good as transferring a discovered subnetwork. In contrast, freezing and not freezing the discovered subnetwork does not have significant differences.

We draw two insights from this control: First, subnetworks discovered through our method are likely *directly reusable*, considering freezing it does not hurt performance like the random control does. Secondly, transferring a randomly sampled subset of weights is likely equivalent to a "better initialization", echoing Frankle & Carbin (2019)'s findings.

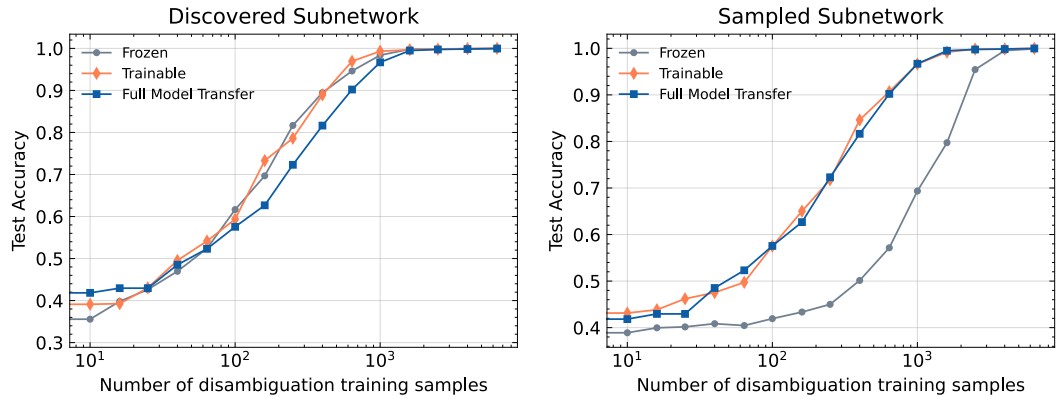

Figure 7: Comparison between Subtask Induction allowing for gradient updates within the subnetwork and frozen subnetwork. Allowing for gradient updates does not improve the performance of Subtask Induction, but makes a significant difference for a randonly sampled subnetwork.

### B.4 ADDITIONAL ARITHMETIC COMPUTATIONS

In addition to the single arithmetic task in the main paper, we experiment with a range of different original and downstream computations and transfer potentially useful subnetworks between these computations. Following Power et al. (2022), we define the following arithmetic computations, where $p = 7$ and range of input tokens is $[0, 1000)$:

$$a + ab \pmod{p} \tag{1}$$
$$a^2 + b^2 \pmod{p} \tag{2}$$
$$a^2 + ab \pmod{p} \tag{3}$$
$$a^2 + ab + b^2 \pmod{p} \tag{4}$$
$$a^3 + ab \pmod{p} \tag{5}$$
$$a^2 - b^2 \pmod{p}. \tag{6}$$

The computation in Equation 1 is used for finding the subnetwork implementing $ab \pmod{p}$ in the main paper. In addition, we use the same setup as those described in Section 4.2 to train for a 12-layer GPT2 model on Equation 2. We then localize a subnetwork performing $a^2 \pmod{p}$ from the original model, which achieves perfect accuracy with 6.79% parameters. Similarly, we train a base model on Equation 3, and find a subnetwork containing 6.37% of masked parameters computing $a + b \pmod{p}$ with perfect accuracy, which is one possible way of solving the task (as the composition of $a$, the $\times$ operator, and $a + b$). This combined with the $ab \pmod{p}$ subnetwork introduced in the main paper gives a total of 3 different subnetworks.

We then experiment with transferring the 3 subnetworks to different downstream computations where the subnetwork is likely beneficial. These experiments are structured as follows:

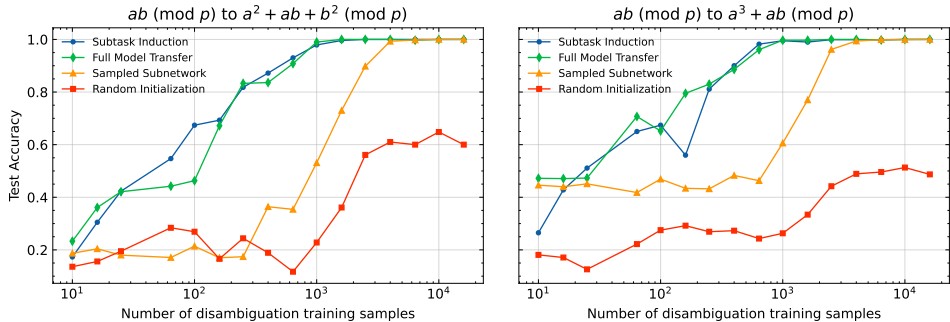

Figure 8: Test accuracy vs. number of training samples for transferring a subnetwork capturing $ab \pmod{p}$ to different downstream computations. We observe the same effects as transferring the subnetwork to $a^2 + ab \pmod{p}$: Subtask Induction yields comparable accuracy compared to transferring the entire model, and performs much better than training from random initialization or a randomly sampled subnetwork.

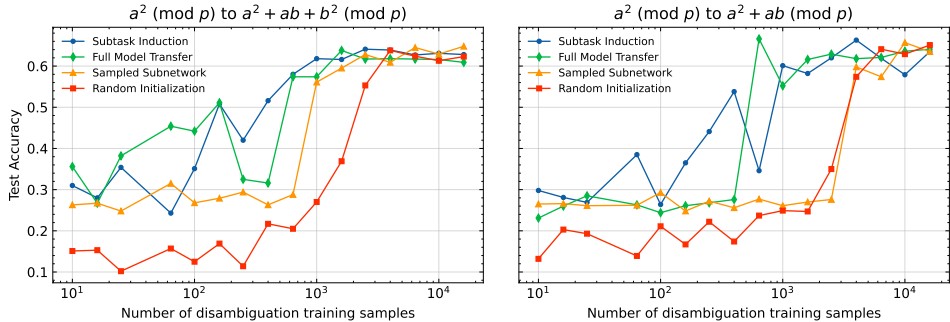

Figure 9: Test accuracy vs. number of training samples for transferring a subnetwork capturing $a^2 \pmod{p}$, found from a base model trained on $a^2 + b^2 \pmod{p}$, to $a^2 + ab + b^2 \pmod{p}$ and $a^2 + ab \pmod{p}$. We note that compared to controls, Subtask Induction performs better, albeit with much less stability and performance gap, at low data regimes. However, this performance and the performance of transferring the entire model are matched by training from scratch when the number of training samples becomes high, which does not occur for the previous set of experiments.

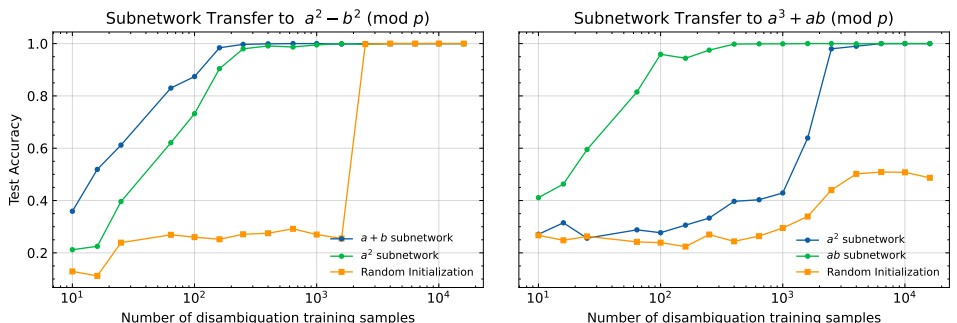

Figure 10: Test accuracy vs. number of training samples for transferring different subnetworks to the same task. Left: we transfer both an $a + b \pmod{p}$ subnetwork and a $a^2 \pmod{p}$ subnetwork to the task $a^2 - b^2 \pmod{p}$, which potentially encourages the model to adopt different solutions. We find that though both subnetworks provide significant benefits, transferring the $a + b \pmod{p}$ subnetwork yields better performance, perhaps suggesting that the model is more biased towards adopting that solution. Right: we transfer both an $ab \pmod{p}$ subnetwork (helpful) and an $a^2 \pmod{p}$ subnetwork (unhelpful) to the computation $a^3 + ab \pmod{p}$, and find that the former brings better transfer performance as expected.

1. We present results transferring the same subnetwork as Section 4, but to different downstream computations, in Figure 8. In Section 4 we transfer a subnetwork computing $ab \pmod p$ to the task $a^2 + ab \pmod p$, which has the subtask captured by the subnetwork as constituent. Since Subtask Induction should be beneficial as long as the transferred subtask is a constituent of the new computation, we additionally define two more complex computations—$a^2 + ab + b^2 \pmod p$ and $a^3 + ab \pmod p$—and transfer the same $ab \pmod p$ subnetwork to these computations. We find similar effects as those observed in Section 4.

2. In 9, we additionally validate Subtask Induction by transferring a different subnetwork, one that computes $a^2 \pmod p$ localized from a base model trained on $a^2 + b^2 \pmod p$, to two computations both having the subtask as their constituent. Even though the general trend is similar, transferring this subnetwork achieves overall worse results with higher variance in performance. One plausible explanation could be the difference in the structure of the base model and downstream models: $a^2 + b^2 \pmod p$ does not have interaction terms between the two inputs, and models may learn a different structure than those needed for the computations the subtask is transferred to. We leave the exploration of when and how such structures emerge as future work.

3. In 10, we evaluate the efficacy of Subtask Induction by transferring different subnetworks that could potentially help solve the new computation, and by transferring subnetworks that are irrelevant to the new computation. The setups are as follows: for the computation $a^2 - b^2 \pmod p$, two equivalent subroutine compositions are possible: breaking down to $a + b$ and $a - b$, and breaking down to $a^2$ and $b^2$. We thus transfer both a $a + b \pmod p$ subnetwork and a $a^2 \pmod p$ subnetwork to this task and compare their performance. For the computation $a^3 + ab \pmod p$, we expect a subnetwork computing $ab$ to be helpful to it, and a subnetwork computing $a^2$ to be irrelevant. We transfer both subnetworks to this task and evaluate their performance, and find that even though transferring an irrelevant subnetwork still provides performance gains compared to random initialization, it is much less effective than transferring computations that are genuinely useful.

## C  EXTENDED VISION RESULTS

### C.1  MODEL CONFIGURATION AND HYPERPARAMETERS

For subnetwork discovery, we initialize masks on every layer excepts the classifier layer and train the mask for 100 epochs with batch size =32, $\beta_{\text{final}} = 100$, and sparsity penalty $\lambda \in \{1.0^{-6}, 1.0^{-7}, 1.0^{-8}\}$. We always transfer the subnetwork with highest accuracy.

Similar to He et al. (2015a), we use SGD optimizer with momentum = 0.9, weight decay = 0.0001, and learning rate $1.0^{-3}$ to train our from-scratch and re-initialized ResNet models. We use a batch size of 128, which we find to be sufficient for both models to converge, and train for a total of 33,160 steps. For ViT-base, we find a batch size of 512 to be optimal. We experiment with both the SGD used for ResNet training runs and Adam with the same setup as original authors (Dosovitskiy et al. (2020), $\beta_1 = 0.9, \beta_2 = 0.999$, weight decay = 0.03) and a similar linear learning rate scheduler. We find SGD to produce better results in our experiments and report these results for Figure 5.

### C.2  RANDOM CONTROL FOR SUBNETWORK DISCOVERY

We run controls for subnetwork discovery on vision models in order to investigate if the probed subnetworks represent emergent structure from pertaining, or if they are simply suitably initialized parameters that also exist in models as initialization. To do this, we apply subnetwork discovery on both pretrained ResNet18 and ViT and their randomly initialized counterparts. As shown in Figure 11, we find that subnetworks trained from randomly initialized models are worse in performance and different in training dynamics compared to those localized from pretrained models. In pretrained models, as an artifact of the annealing process and temperature scheduling, we often see no increase in subnetwork performance for a significant part of the early training process. This does not seem to be the case for subnetworks found from randomly initialized models. Furthermore, the found subnetworks from randomly initialized models only achieve around 35% accuracy on Mean-pooled ImageNet in both ResNet18 and ViT. Since these accuracies are uniformly worse than those of the entire pretrained model, it is unlikely that they can help instill effective biases.

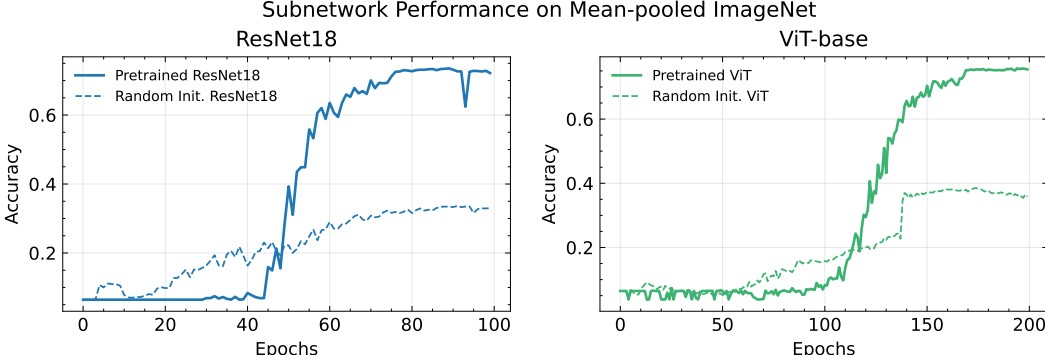

Figure 11: Training dynamics of *subnetwork discovery* for vision models, comparison between pretrained models and randomly initialized ones. Left: evaluation accuracy vs. epochs for ResNet18; right: evaluation accuracy vs. epochs for ViT. We see that subnetworks localized from pretrained models significantly outperform the entire pretrained model on Mean-pooled ImageNet, while subnetworks localized from randomly initialized models are much worse in performance.

### C.3 RESULT BREAKDOWN BY CLASS

In addition to averaged results, we additionally report result breakdown by class for both pretrained ResNet18 and ViT-base and our Subtask Induction models. We note that all pretrained models perform less optimally on classes where textures are more dominant (e.g. bears, cats, dogs, and elephants), and models trained with Subtask Induction generally perform better at these classes. However, all 4 models also seem to struggle with mean-pooled images of bicycles. This could potentially due to the shape of bicycle frames not being apparent in mean-pooled images. We also find that our models generally struggle with the class "knife". We leave further exploration of these findings to future work.

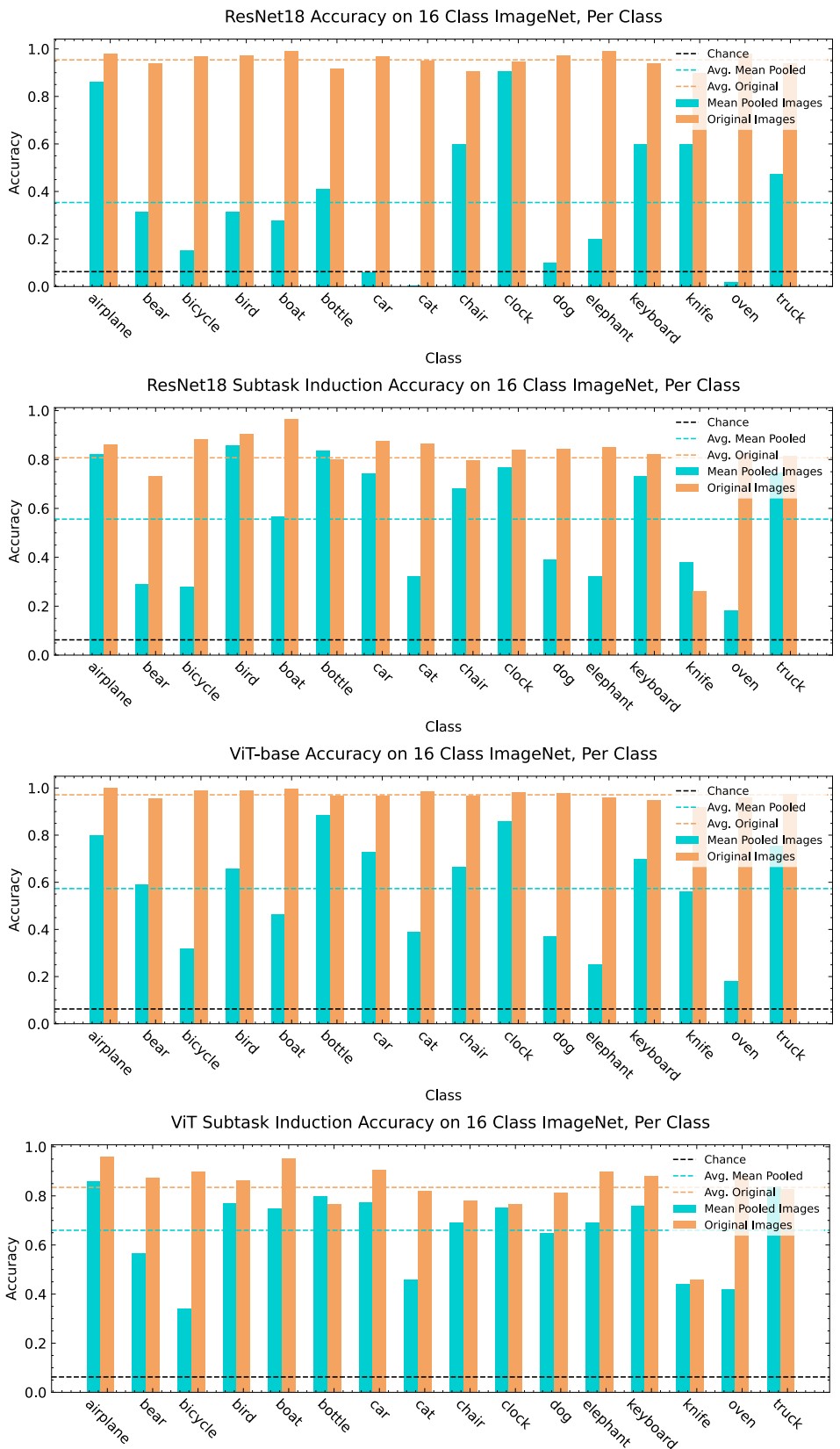

