# OpenReview forum: "Instilling Inductive Biases with Subnetworks"
_ICLR.cc/2024/Conference — Submitted to ICLR 2024_

### Official Review · Reviewer_wyEr · 2023-10-29

**Soundness:** 3 good
**Presentation:** 4 excellent
**Contribution:** 3 good
**Rating:** 6
**Confidence:** 3

**Summary:**

This paper discusses the problem of injecting inductive bias into a network, via the proposed subtask induction method. In particular, using an auxiliary subtask dataset, the authors first identify a subnetwork that achieves the subtask and use it as the initialization for another network, which is subsequently trained for a closely related downstream task. To show the effectiveness of the method, this paper conducted two sets of experiments: an arithmetic computation with decoder-only transformers, and a vision experiment.

**Strengths:**

Overall the problem statement is quite interesting and the presentation is very clear and easy to follow.

**Weaknesses:**

I found the motivation to be a little lacking. The authors mentioned mechanistic interpretability, however, it is unclear to me (based on the current writing) why we should care about such interpretability. I would encourage the authors to give some examples to demonstrate how it can be used in reality. For example, what are some scenarios in which we can accurately define subtasks (in the vision experiment you mention shape vs texture, I understand that the problem is extensively studied, but it is also kind of artificial)?

**Questions:**

1. For the GPT2 experiment, did you use their pre-trained weights, or did you train everything from scratch?
2. If you tokenize the integers from {0, ..., 999} with GPT2tokenizer, 181 of them are actually tokenized into 2 tokens (for example, 521->'5', '21'). If the original tokenizer is used, I would suspect the model cannot really learn the underlying equation (this may also be why you need many disambiguous examples).
3. How does the number of ambiguous examples affect the accuracy? Does the performance decrease with more examples?
4. In the first paragraph in section 5.3, to confirm, you discover pretrained ResNet18 and ViT, are they pretrained on only ImageNet? If you have a randomly initialized ViT/RN, can you still discover a subnetwork that works well on Mean-pooled ImageNet?
5. What is the training setup of Figure 5? Did you train every model on 213k images of the 16-class ImageNet? The comparison looks a little unfair to me, as the Subtask Induction models have been pretrained (maybe on the original ImageNet? This question is related to the previous point). A more fair comparison will be that you find a subnetwork from a randomly initialized network that performs well on the mean-pooled imagenet.
6. In the right figure in figure 5, the accuracies don't change.  Is this because you froze the subnetwork's weights after transferring? If you don't freeze the weights, will the accuracies on mean-pooled imagenet increase?
7. Can you also give the detailed setup of each row in Table1? My current understanding is: model+subtask induction = model pretrained on all original ImageNet, find subnetwork using the mean-pooled data, then continually train using 213k 16-class imagenet. model from scratch is trained to perform 16-class classification directly. model+DA and model+pretrained are where I get confused: a) how exactly is the data augmentation performed? and b) is "model+pretrained" where you finetune only the last classification layer? If so, how exactly is the last-layer finetuned?

miscellaneous: table caption should be on top; paragraph on top of section 4.3 'diambiguation'->'disambiguation'.

I'm happy to raise my rating if the authors can address my concerns.

---

> ### Author Response · Authors · 2023-11-21
> **Response 1/2**
>
> Thank you for your thoughtful comments and questions! For motivation, we see Subtask Induction as a way to instill custom inductive biases, which are often thought to mediate faster learning and better robustness in both ML systems and also human learning. We mention in our introduction that we often want these inductive biases because (1) they enable learning with much less data and (2) more human-aligned biases likely result in better OOD generalization. In the vision experiment, we use Subtask Induction to encourage the model to adopt a more human-like solution that is otherwise not preferred, mitigating potential model heuristics. There are lots of real-world cases where the complete solution to a task is hard to define, but we have knowledge to a subtask that potentially helps to solve the task (e.g. for RL agents, this is often in the form of sets of basic actions; for language models, a syntax parser or domain-specific knowledge may be especially helpful for many tasks). Traditional approaches to this problem is usually data augmentation, and we see Subtask Induction as providing a more efficient and reliable alternative that mechanistically localizes a subtask instead of providing soft biases through data. Another reason one might want to use Subtask Induction is because of the complex nature of many machine learning tasks: hand-designing neurosymbolic architectures requires much more extensive domain knowledge and limits the model's capabilities. By transferring a subnetwork, we give the model some useful information without limiting its ability to learn the rest of the solution. In our current paper we introduce the approach and give a validating set of experiments. However, we are highly hopeful for broader applications, such as the examples mentioned above, and are excited to explore them as follow-ups.
>
> Below we respond to each of the questions:
>
> - **Q:** For the GPT2 experiment, did you use their pre-trained weights, or did you train everything from scratch?
> **A:** We trained everything from scratch. This is because we experiment with much smaller models (2-12 layers, 128 embedding size), and much more architecture variations. In addition, the task we are training the models for is purely algorithmic, so it's very unlikely that using a pretrained model will help.
> - **Q:** If you tokenize the integers from {0, ..., 999} with GPT2tokenizer, 181 of them are actually tokenized into 2 tokens (for example, 521->'5', '21'). If the original tokenizer is used, I would suspect the model cannot really learn the underlying equation (this may also be why you need many disambiguous examples).
> **A:** We tokenize each numerical input as an individual token (e.g. 997 and 998 are two different tokens, each with its own embedding, and there are 1000 tokens for all the possible inputs), thus the model has no prior information about the relationship between numbers. This arithmetic task is thus more like a large 1000x1000 binary operation table that follows some specific rules. We detail this setup in Section 4.1. This design is in line with the paper that introduced this task [(Power et al.)](https://arxiv.org/abs/2201.02177), and in Figure 5 (page 8) of the linked paper the original authors present a visualization for this operation---since models are presented with numbers with no internal structure (such as being composed of digits), it's quite impressive that models learn this task perfectly without being presented all possible combinations.
> - **Q:** How does the number of ambiguous examples affect the accuracy? Does the performance decrease with more examples?
> **A:** Since ambiguous samples are merely ambiguous (they still provide the right solution to the task), having more ambiguous samples will likely only help. This is because these samples enforce that each of the possible inputs appear at least once, even in cases where the total number of training samples is less than 1000. This helps train embeddings for each of the tokens.

---

> ### Author Response · Authors · 2023-11-21
> **Response 2/2**
>
> - **Q:** In the first paragraph in section 5.3, to confirm, you discover pretrained ResNet18 and ViT, are they pretrained on only ImageNet? If you have a randomly initialized ViT/RN, can you still discover a subnetwork that works well on Mean-pooled ImageNet?
> **A:** Yes, we use ResNet18 and ViT-base from original authors, which are pretrained on ImageNet. As binary masking severely limits the expressitivity of the model, it's quite unlikely that we can find a well-performing network from random initialization. We added an experiment where we trained for subnetworks using the same setup for ResNet and ViT, but from randomly initialized models. Though we still find higher-than-chance accuracy (around 35% for both models), the performance of these subnetworks are much worse than ones found from pretrained models. We have included an additional section in the appendix (Appendix C.2, pg. 18-19) for a training dynamic comparison on subnetwork discovery on pretrained vs. randomly initialized models.
> - **Q:** What is the training setup of Figure 5? Did you train every model on 213k images of the 16-class ImageNet? The comparison looks a little unfair to me, as the Subtask Induction models have been pretrained (maybe on the original ImageNet? This question is related to the previous point). A more fair comparison will be that you find a subnetwork from a randomly initialized network that performs well on the mean-pooled imagenet.
> **A:** Yes, we train every model on the 213k images of 16-class ImageNet. We agree that the subnetwork transfers partial knowledge from its upstream model, and its raw accuracy should not be compared independent of this context. However, it's worth noting that these models achieve and retain accuracies on mean-pooled images that are not just higher than from-scratch models, but also pretrained models that has been trained on magnitudes more data. As we added to the Appendix as random control, such subnetworks do not exist in randomly initialized models.
> - **Q:** In the right figure in figure 5, the accuracies don't change. Is this because you froze the subnetwork's weights after transferring? If you don't freeze the weights, will the accuracies on mean-pooled imagenet increase?
> **A:** Yes, this is likely because we froze the subnetworks. However, it's unlikely that unfreezing the subnetwork will lead to better performance because (1) our training data does not contain mean-pooled images, and the accuracy of our models on mean-pooled images already outperform those of the entire pretrained models, and (2) if we mix in the 15.4k images used for probing, it will likely only be detrimental to model performance as it leads to overfitting, as we demonstrated in "Data Mixture" control in Table 1 (which fine-tunes the pretrained model with these mean-pooled images, and find that mean-pooled accuracy actually *decreases* compared to if we just use the pretrained model and adapt the classification head).
> - **Q:** Can you also give the detailed setup of each row in Table1? My current understanding is: model+subtask induction = model pretrained on all original ImageNet, find subnetwork using the mean-pooled data, then continually train using 213k 16-class imagenet. model from scratch is trained to perform 16-class classification directly. model+DA and model+pretrained are where I get confused: a) how exactly is the data augmentation performed? and b) is "model+pretrained" where you finetune only the last classification layer? If so, how exactly is the last-layer finetuned?
> **A:** Sorry for the confusion! We realized that we did not sufficiently specify our setups, and used names that might be misleading. We have changed "data augmentation" to "data mixture". For this control, we fine-tune a pretrained base model on a mixture of both Mean-pooled ImageNet and original ImageNet images, with the hope of instilling a bias towards shape-based classfication. Since these images share the same output space, we simply mix all images during fine-tuning. For pretrained baselines, we re-initialize a classification head with 16 output classes and train only that head while keeping the base model frozen on all 213k images of 16-class ImageNet. Even though 16-class ImageNet has classes aggregated from original ImageNet classes, [Geirhos et al.](https://openreview.net/forum?id=Bygh9j09KX) reported that retraining instead of taking aggregated probabilities work better, so we adopt this strategy. In practice we see that training the new classification head converges quickly (within 1 epoch), and further training does not yield any significant difference.
>
>
> We moved our table caption and fixed the typo in Section 4.3. Thanks for pointing those out!

---

> > ### Comment · Reviewer_wyEr · 2023-11-22
> > **Thanks for your response**
> >
> > Why is RN18+Data Mixture trained on 1.28M+15.4K? If the model has a head of 16 classes, then shouldn't it be trained on 213k+15.4k? Also how exactly is the data sampled from this data mixture? Since the data is quite unbalanced, in each batch, do you enforce roughly the same number of mean-pool images and original images?

---

> > > ### Author Response · Authors · 2023-11-22
> > >
> > > This is because we directly fine-tune the base model, which has been trained on all 1.28M ImageNet-1k images. This is similar to how ViT counts their training data (ViT is pretrained on 14M of ImageNet-21k, then fine-tuned on 1.28M of ImageNet-1k). We experiment with both no upsampling (so the dataset remains unbalanced) and upsampling mean-pooled images to 1:1 with original images and find similar performance. The one reported in the table is with no upsampling.

---

> > > > ### Comment · Reviewer_wyEr · 2023-11-23
> > > >
> > > > So correct me if I am wrong. 1.28M+15.4k means 1) the model is pretrained on 1.28M, and subsequently 2) finetuned on 16-class 213k imagenet+15.4k mean-pool images without upsampling. Do you have the result somewhere in the paper that uses upsampling?

---

> > > > > ### Author Response · Authors · 2023-11-23
> > > > >
> > > > > Yes, that's correct. We do not currently have these in the paper as the previous reply describes an initial experiment we did while doing controls. We will run a more in-depth range of these experiments for both ResNet and ViT with varying data mixture ratios and include these in the final manuscript. Thank you for the follow-up!

---

### Official Review · Reviewer_pnkg · 2023-10-31

**Soundness:** 2 fair
**Presentation:** 4 excellent
**Contribution:** 2 fair
**Rating:** 6
**Confidence:** 3

**Summary:**

The authors propose Subtask Induction, a way of implanting inductive bias by identifying a subnetwork that performs a specific task, and only training the rest of the model. The idea is that the rest of the model is forced to learn, bounded to performing that specific (wanted) task. The method is tested on two problems. The first one is the reconstruction of a discrete mathematical operation, and the second is image classification on 16-class ImageNet. In both cases Subtask Induction obtains better perfomances in situations where the inductive bias was necessary and that were hard to learn for a model where this was not instilled.

I lean towards acceptance because:
- The idea of inducing a specific inductive bias is intriguing and deserves to be discusses in a wider context. Similarly to transfer learning, one could imagine collections of pretrained subnetworks that are specifically tailored towards a task, or that avoid some specific spurious correlation.
- The paper is well-written and easy to follow, and the code is well-documented.

I score the paper as a 6, which I reserve myself to lower if some of the answers to my questions are not satisfactory or if some other reviewer.

The reason for not giving a higher score is because though the idea is nice, (i) it is hard to visualize how this method could be actually implemented efficiently and (ii) the evidence is purely empirical, and the set of experiments is not very extensive. The reason why it is not too extensive is that it is hard to devise suitable testing situations, which sends me again to point (i).

**Strengths:**

- The method can potentially help solving, in the long run, problems related to spurious correlations, out-of-domain generalization, and more.

- A well-documented code is made available through an anonymous link.

**Weaknesses:**

- The method is valid for one subtask, but extending it to more than one is non trivial and perhaps not possible, since the subnetworks could either overlap or occupy the whole network (in which case we would just be doing transfer learning).

- It is not easy to identify the subnetwork of a model that performs a subtask. I see this as a major limitation, but it doesn't seem impossible to me that in the future there could be some solutions.

- The method relies on finding a subnetwork, which seems a very intensive work.

- The method is validated on a limited number of tests. If on one side it is hard to find real-life examples (the authors had to resort to an adhoc dataset), they could at least have tried with some other mathematical operations (and there, architectures since in principle the method should apply to any architecture, even e.g. MLPs).

**Questions:**

- Can the authors elaborate on the difference between their method and transfer learning? The two seem very similar to me, and transfer learning would be the limit for very big subnetworks.

- In transfer learning, we fine tune by first freezing the model and training only the last layer, and then training the whole model with a very small learning rate. Here, the second step was deliberately removed in order to enforce the subtask. Could a second step of fine tuning help?

- Is the dataset available at this point? I cannot find where it is written.

- Could it be possible to benchmark this method on datasets for spurious correlations?

- What do the found subnetworks look like? Are they restricted to a single layer? Are they dense blobs? How many paths do they comprise?

- How does the method deal with dropout? With dropout there are no (or fewer) preferential paths, so I would assume the subnetworks would be much bigger.

- The subnetwork can only be transferred without varying the architecture, right? E.g. there cannot be a subnetwork which is only made of two layers, and I plug these two layers into my model. Is this right? Or, if I understood wrong, how is the subnetwork transferred when the architecture changes?


Other minor things:
- Figure 1 is not referenced in the text
- typo on page 13: implementatione
- In section 3.1/2 it is sort of obvious that the masks are trained on well-trained models (instead of e.g. models at initialization) but it is not explicitly written
- I would put a parenthesis before the modulo operators, because I was initially confused. For example, I would write

---

> ### Author Response · Authors · 2023-11-21
> **Response 1/3**
>
> Thank you for your thoughtful comments and feedback! We agree that Subtask Induction in its current iteration still has lots of room for improvement. In our work we hoped to establish that it is possible to find subnetworks, and that these subnetworks can be used to instill custom inductive biases. We're excited by this initial empirical success, and hope to explore in future both more in-depth explanations for this phenomenon and methods for more effectively localizing & transferring subnetworks. We believe combining our approach with insights on biases of models & datasets that identify them can pave the way to instilling more desirable model behaviors with higher data efficiency. Below we answer each of the questions/concerns:
>
> **Weaknesses**
>
> - **Q:** The method is valid for one subtask, but extending it to more than one is non trivial and perhaps not possible, since the subnetworks could either overlap or occupy the whole network (in which case we would just be doing transfer learning).
> **A:** While subnetworks may overlap, we do not see it as making transferring both subnetworks impossible a priori. Building on the intuition that a subnetwork captures useful representations for the subtask, and other parameters are trained "around" it to leverage these representations, transferring multiple subnetworks as their union will give model information from all subnetworks as long as the two subnetworks are derived from the same base model, and thus share parameters. We see this as a potential direction for future work to instill more nuanced inductive biases. For the case the subnetworks occupy the entire network, please see our response to Question 7 (transferring with varying architecture)---moreover, if all the behaviors from an original model is desirable, we may as well be justified to do transfer learning.
> - **Q:** It is not easy to identify the subnetwork of a model that performs a subtask. I see this as a major limitation, but it doesn't seem impossible to me that in the future there could be some solutions.
> **A:** Indeed, Subtask Induction requires that we first localize a desirable subnetwork. While the current iteration is already quite data-efficient, we hope future work can further simplify this step.
> - **Q:** The method relies on finding a subnetwork, which seems a very intensive work.
> **A:** Finding subnetworks is in itself a training process, which in its current form is indeed quite intensive. We hope to explore ways to locate subnetworks more effectively in the future, perhaps by changing the scheduled annealing routine or employing other approaches to sparsification. Since training binary masks on top of original parameters severely limits the search space of the training process, and as a result is very data-efficient (we only used 15.4k images for identifying subnetworks in ResNet and ViT), further improvements should be possible. With our current approach, the training process turns out to be around 1-3 hours on a single 3090.
> - **Q:** The method is validated on a limited number of tests. If on one side it is hard to find real-life examples (the authors had to resort to an adhoc dataset), they could at least have tried with some other mathematical operations (and there, architectures since in principle the method should apply to any architecture, even e.g. MLPs).
> **A:** Thank you for the suggestion! We share the reviewer's opinion that validation on multiple tasks/architectures is important. We have updated our manuscript (Appendix B.4 on pg. 16-18) to include more experiments. These experiments include transferring the same $a + b$ (mod $p$) subnetwork to different computations, transferring a different $a^2$ (mod $p$) subnetwork, and a controlled experiment transferring different subnetworks to the same computation. We find converging results that are in line with the observations we made in the main paper, and thus put these in the appendix. We believe these additional experiments give both more empirical evidence of the efficacy of our method, and provide better insights on when they work/fail (experiments presented in Fig.10). For architecture, we have worked with both encoder-only(ViT) and decoder-only(GPT2) transformers and CNNs(ResNets), which are relevant and practically useful architectures. We believe this range of models & tasks gives adequate evidence that Subtask Induction is at least somewhat general across model architectures.

---

> ### Author Response · Authors · 2023-11-21
> **Response 2/3**
>
> **Questions:**
>
> - **Q:** Can the authors elaborate on the difference between their method and transfer learning? The two seem very similar to me, and transfer learning would be the limit for very big subnetworks.
> **A:** Our method shares the intuition with transfer learning that pretrained models possess representations helpful for downstream tasks. However, even large pretrained models learn heuristics that are hard to mitigate (see [Geirhos et al.](https://openreview.net/forum?id=Bygh9j09KX) for CNNs, and [McCoy et al.](https://arxiv.org/abs/1902.01007) for a similar case in BERT), and transfer learning simply transfers the bias downstream. Our method leverages on two key insights: (1) models often capture desirable subtasks even if they are generally more reliant on other cues and (2) if a model is trained from subtasks, it becomes more likely to rely on the subtask. In our experiments, we show that models trained with the same data distribution overcome common heuristics for CNNs. If we wish to transfer all information from a pretrained model, we may simply do transfer learning. However, in many cases if there's an identifiable heuristic we wish to avoid, Subtask Induction provides a method for extracting it that's more data- and computation-efficient compared to re-training base models with a different data regimen.
>
> - **Q:** In transfer learning, we fine tune by first freezing the model and training only the last layer, and then training the whole model with a very small learning rate. Here, the second step was deliberately removed in order to enforce the subtask. Could a second step of fine tuning help?
> **A:** In our arithmetic experiments, we experiment with performing subnetwork transfer both freezing & not freezing the subnetwork (Appendix B.3), and find negligible difference between these approaches. We thus made the assumption that based on the model's preference against shape-based cues, it is unlikely that allowing for gradient updates within the subnetwork would help. We're happy to run similar controls for vision tasks---is this what you mean by "second step of finetuning"?
> - **Q:** Is the dataset available at this point? I cannot find where it is written.
> **A:** Due to the size of the complete dataset it's quite hard to host it anonymously. We plan to host it on Huggingface publicly and link it from our Github repository.
> - **Q:** Could it be possible to benchmark this method on datasets for spurious correlations?
> **A:** In Section 5.4 (cue conflict analysis), we analyze one such case by testing on a cue conflict dataset [(accessible online here)](https://github.com/rgeirhos/texture-vs-shape/tree/master/stimuli/style-transfer-preprocessed-512) that computes style transfer from one class to another, and our model demonstrates better robustness than vanilla ResNet on these images. We are interested in exploring more datasets that help identify spurious correlations and see if Subtask Induction proves to be helpful in these scenarios. Do you have any suggestions?
> - **Q:** What do the found subnetworks look like? Are they restricted to a single layer? Are they dense blobs? How many paths do they comprise?
> **A:** Based on our initial observations, subnetworks are often distributed across heads and layers, though patterns across earlier and latter layers are often quite different (e.g. we have seen some subnetworks to be more distributed across weights in initial layers, and more concentrated in neurons in latter ones). In ResNets, we often observe symmetry in convolution filters (the mask is sometimes roughly symmetric across the center, or w.r.t some axis). Since all modern deep networks have residual connection, it's hard to count the number of "paths" as even if all subsequent weights are masked out, residual connection still helps the activations at some intermediate layer reach the final output. We will add some visualization and potentially analyses to our appendix.

---

> ### Author Response · Authors · 2023-11-21
> **Response 3/3**
>
> - **Q:** How does the method deal with dropout? With dropout there are no (or fewer) preferential paths, so I would assume the subnetworks would be much bigger.
> **A:** Subtask Induction does not apply special treatment for dropout. For all GPT2 experiments, we use the default attention, embedding, and feed-forward dropouts of 0.1. We train the binary mask with the base model in evaluation mode, thus disabling dropout, and obtain relatively small subnetworks (Table 2, generally below 10%). ResNet does not have dropout and ViT does, and we find subnetworks of around 15% of total parameters in both models. For the same model trained on the same task, we do expect models with higher dropout will have larger subnetwork size. But since information processing might already be somewhat distributed in neural networks, we do not observe a significant differences in subnetwork size/performance across these settings.
> - **Q:** The subnetwork can only be transferred without varying the architecture, right? E.g. there cannot be a subnetwork which is only made of two layers, and I plug these two layers into my model. Is this right? Or, if I understood wrong, how is the subnetwork transferred when the architecture changes?
> **A:** Currently, yes. A recent work ([LiGO](https://vita-group.github.io/LiGO/)) has proposed a method to extend the depth and width of transformers and has shown that such extended transformers enable learning on top of the original model. Since the main insight in Subtask Induction is that gradient descent allows models to learn based on subnetworks, it seems possible that it can be combined with LiGO to at least expand upon the same architecture, thereby allowing for reuse across different sized models and expanding model capabilities. We envision this to be important for more complex applications and are excited to explore this as future work. We have updated our future work section to include this discussion.
>
> For minor comments, our submission has been updated to reflect these changes:
>
> - **Figure 1:** Figure 1 is now introduced in the second paragraph of Section 3.
> - **Typo:** fixed.
> - **Section 3.1/3.2:** In the second paragraph of Section 3 as well as the first sentence of 3.1, we mention that subnetwork discovery is performed on a trained neural network.
> - **Presentation of modulo:** For consistency, we use the same notation as prior work ([Power et al.](https://arxiv.org/abs/2201.02177), [Nanda et al.](https://arxiv.org/abs/2301.05217)). We also updated the second paragraph in Section 4.1 (where the task is introduced) to explicitly state that the modulo is taken after original operations.

---

### Official Review · Reviewer_PuHj · 2023-11-03

**Soundness:** 4 excellent
**Presentation:** 4 excellent
**Contribution:** 3 good
**Rating:** 8
**Confidence:** 3

**Summary:**

The authors present a new approach called Subtask Induction, which involves extracting specific problem-solving abilities from trained neural networks. They achieve this by isolating a subnetwork responsible for a particular task within a larger neural network and initializing another network with only these subnetwork weights, leaving the rest randomly initialized. The authors demonstrate subtask induction on an arithmetic task and image classification on a novel dataset, Mean-pooled ImageNet, which requires networks to learn shape information rather than texture information. The approach is an interesting demonstration that (1) these subnetworks exist and can be extracted and transferred to new networks, (2) SGD learns to use these subnetworks, and this leads to more efficient learning on new tasks.

**Strengths:**

I really enjoyed this paper - it takes the observation from the mechanistic interpretability literature that deep networks learn subnetworks to solve specific tasks and uses it to derive a simple method for extracting these subnetworks (essentially by optimizing a sparsity mask over the parameter on a distribution of problems that only require the subnetwork) and then they randomly initialize the remaining weights of a network and train on a second task that requires the shared skill. The results on the arithmetic task provide a proof of concept, but vision experiments are particularly interesting because they show that these subnetworks can be discovered in real data, provided you have dataset which allows you to extract this.

**Weaknesses:**

The requirement for a dataset like mean-pooled imagenet to extract the subnetwork significantly constrains how widely applicable this paper is as a method---it essentially requires you know the task in advance and how to specify it with examples that are sufficiently different from typical examples in the training set---but I still think that it is an interesting demonstration.

I would have liked to see some examples of where it fails: for example, if you don't have a clear separation between IID samples and the target task, I would expect it would struggle.

**Questions:**

1. Did you study any tasks for which Subtask Induction performs poorly?
2. If so, what are the characteristics of those tasks?

---

> ### Author Response · Authors · 2023-11-21
>
> Thank you for your positive review and insightful question! We agree that having to design a dataset and localize the subnetwork through a supervised training process is limiting. While in many cases the dataset is not impossible to define (i.e. for our case we performed custom data augmentation, or for language-related tasks, one might want a dataset that captures certain syntactical structures or behaviors), it does take work to construct and validate it. If such dataset cannot be effectively separated from normal training data, it is likely that we obtain a subnetwork that performs whatever it takes to solve *that dataset* instead of the desired subtask---in the extreme case, this would just mean pruning the original model (assuming the subtask is completely inseparable). Our intuition is that transferring this "subtask" would certainly still help boost performance, but may not help instill desirable biases.
>
> We have added an additional experiment where we specifically explore cases where we hypothesize Subtask Induction to either fail or perform poorly on. These experiments, based on our arithmetic experiments, transfer different subnetworks for the same downstream computation. In the first case, we transfer an $a + b$ subnetwork and an $a^2$ subnetwork to the computation $a^2 - b^2$ (mod $p$), where we hypothesize that the first encourage the model to learn $(a + b)(a - b)$, and the second encourages breaking down $a$ and $b$ inputs. We find that---to our surprise---though both subtasks help, transferring $a + b$ gives better performance, despite more difficulty in composing the subtask into the training objective. This is one case where Subtask Induction may contradict human intuitions on what helps. In another case, we transfer $a^2$ and $ab$ subnetworks to $a^3 + ab$ (mod $p$), where the first subnetwork is expected to be somewhat useless, and the second should be helpful. Expectedly, transferring $ab$ in this case brings much larger performance gains than transferring $a^2$, but even the $a^2$ subnetwork still provides benefits compared to training from random initialization. These results are presented in pg. 16-17 in Appendix B.4.

---

### Author Response · Authors · 2023-11-21

We thank the reviewers for their reviews and feedback! In addition to responding to concerns & questions from each reviewer individually, we summarize key changes made to the manuscript below.

**Change Log:**

- Added an additional 3 sets of experiments to Appendix B.4, demonstrating the effect of Subtask Induction on different computations & subnetworks. These experiments also give examples of subnetworks that have varying performances when transferred to downstream tasks.
- Added a control for vision experiments in Appendix C.2, which trains subnetworks on pretrained vs. randomly initialized ResNet18 and ViT, and find that shape-based subnetworks exist only in pretrained models.
- Updated Table 1 "Data Aug." to "Data Mixture" for clarity. In training these models, we use a mixture of original ImageNet images and Mean-pooled ImageNet images as a baseline of general fine-tuning strategies for adapting to out-of-domain datasets.
- Updated description in Section 4.1 (Dataset for Arithmetic Experiments) to make clear that modulo is taken after the computation.
- Updated future work section to include discussion of transferring subnetworks between models of different sizes.
- Typo corrections and stylistic changes throughout.

---

### Meta-Review · Area_Chair_mKpZ · 2023-12-10

**Metareview:**

This submission presents a technique for instilling inductive biases of a trained model into a new model, a technique that is termed "subtask induction." Reviewers were positive on this submission, finding the idea interesting and the presentation clear, even if the scope of the demonstration was limited to two simple tasks, including one vision task specifically constructed to explore the proposed method.

Because the reviewer confidence was middling, the Area Chair (AC) took a detailed look into this paper at the level of an additional review. The AC had two major concerns. These are:

**(C1): Lack of contextualization and comparison to prior methods to instill inductive biases.**

Subtask induction is, in essence, a technique to adapt a source model to a target domain in a parameter-efficient manner, an idea that is foundational and well-studied in machine learning under many guises: distillation ([Hinton et al., 2013](https://arxiv.org/abs/1503.02531), [Urban et al., 2015](https://arxiv.org/abs/1603.05691)); parameter-efficient transfer learning ([Houlsby et al., 2019](https://arxiv.org/abs/1902.00751), [Iofinova et al., 2023](https://arxiv.org/abs/2111.13445)). Also related is the idea of superposition ([Cheung et al., 2021](https://arxiv.org/abs/1902.05522)) for combining (sub)task-specific models without costly retraining.
However, techniques like these, in particular, transfer methods that are more sophisticated than full fine-tuning, are not cited nor compared to in the present work.

**(C2): Evaluations do not reveal qualitative insight into inductive bias.**

The comparisons between "subtask induction," "from scratch," and "pretrained" reveal that there is a benefit to "subtask induction" on the specific downstream tasks investigated (arithmetic subtask; mean-pooled ImageNet).
We may interpret this as a domain shift penalty incurred by the pre-trained model (because, for example, mean-pooled ImageNet is significantly different from the pre-trained model's pre-training dataset).

The trade-off between leveraging a source model versus training from scratch, and how this interacts with the degree of domain shift, is studied, in greater detail than here, in prior work ([Neyshabur et al., 2020](https://arxiv.org/abs/2008.11687), [Ash & Adams, 2020](https://arxiv.org/abs/1910.08475)).
Importantly, the evaluation (accuracy on the downstream task) in this submission does not provide further insight into the precise qualitative differences in inductive biases between the "subtask induction" models and the "pretrained" models, though this seems to be a major goal of the work (from the abstract and introduction), beyond the brief cue-conflict evaluation inherited from prior work that reveals slight variations in shape bias ([Geirhos et al., 2019](https://arxiv.org/abs/1811.12231)).


In addition, there is no human evaluation on mean-pooled ImageNet, so it is not clear how to interpret the absolute accuracies, nor what a certain threshold of performance reveals about the inductive bias underlying a strategy that achieves that performance level

---

Together (C1) and (C2) imply that this paper presents a technique that is not adequately compared to existing methods for transfer and that the work does not provide significant qualitative insight into inductive bias of large-scale models. As a consequence, the AC recommends that the paper is not ready for acceptance in its current form.

**Justification For Why Not Higher Score:**

insufficient discussion and comparison of relevant prior work (distillation, transfer learning); significance of conclusions wrt the expressed motivation of the work

**Justification For Why Not Lower Score:**

N/A

---

### Decision · Program_Chairs · 2024-01-16

Reject